# Exogenic origin for the volatiles sampled by the Lunar CRater Observation and Sensing Satellite impact

K. E. Mandt [1✉], O. Mousis[2], D. Hurley[1], A. Bouquet[2,3], K. D. Retherford [4,5], L. O. Magaña[4,5] & A. Luspay-Kuti [1]

Returning humans to the Moon presents an unprecedented opportunity to determine the origin of volatiles stored in the permanently shaded regions (PSRs), which trace the history of lunar volcanic activity, solar wind surface chemistry, and volatile delivery to the Earth and Moon through impacts of comets, asteroids, and micrometeoroids. So far, the source of the volatiles sampled by the Lunar Crater Observation and Sensing Satellite (LCROSS) plume has remained undetermined. We show here that the source could not be volcanic outgassing and the composition is best explained by cometary impacts. Ruling out a volcanic source means that volatiles in the top 1-3 meters of the Cabeus PSR regolith may be younger than the latest volcanic outgassing event (~1 billion years ago; Gya).

[1] Johns Hopkins Applied Physics Laboratory, 11100 Johns Hopkins Rd, 20723 Laurel, MD, USA. [2] Aix Marseille Université, CNRS, CNES, LAM, Marseille, France. [3] Aix Marseille Université, CNRS, PIIM, Marseille, France. [4] Southwest Research Institute, San Antonio, TX, USA. [5] University of Texas at San Antonio, San Antonio, TX, USA. ✉email: Kathleen.Mandt@jhuapl.edu

The Lunar Crater Observation and Sensing Satellite (LCROSS) experiment impacted the upper stage of a spent Centaur rocket into the PSR of Cabeus crater, creating a plume that contained the first carbon-, nitrogen-, and sulfur-bearing volatiles detected in the lunar PSRs ([1–3], See Supplementary Table S1). These ground-breaking observations not only provide ground truth for ongoing remote observations of water on the surface (e.g., refs. [4,5]) and at depth (e.g., refs. [6,7]), but provide vital clues to the origin of volatiles present on the Moon. The LCROSS plume was observed 30 s after impact by the Lunar Reconnaissance Orbiter (LRO) Lyman Alpha Mapping Project (LAMP), which detected $H_2$ and $CO$[2,3]. Meanwhile, the LCROSS shepherding spacecraft measured the abundance of several additional species relative to water for 4 min until it also impacted into Cabeus crater[1]. The published abundances from LAMP[2,3] were derived from the expanding shell of vapor traveling at 3–4 km/s that passed LRO when the shell was >100 km away from the impact site. In contrast, the published abundances from the LCROSS shepherding spacecraft[1] were derived from vapor emanating from the impact site over time. Thus, the published LAMP observations were not made at the same time as the LCROSS measurements and require reanalysis for proper comparison (see Supplementary Discussion).

To determine the origin of the volatiles observed in the LCROSS plume we must consider how volatile composition changed between the source, storage in the PSR, and release into the plume. Several processes occur between initial delivery by the source and detection in the plume that change the molecular composition. This means that species that were measured in the plume may not be the same as the molecular species found in the source.

In this work, we simplify the analysis and eliminate as many influences as possible. Instead of using molecular composition we compare the elemental composition of the LCROSS volatiles with the elemental composition of the potential sources, evaluating abundances of four elements as they relate to carbon: hydrogen (C/H), nitrogen (N/C), oxygen (O/C), and sulfur (C/S). Through this analysis we determine that the volatiles sampled by LCROSS are not volcanic in origin, and are most likely cometary.

## Results

### Elemental composition

The elemental composition of the volatiles in the regolith of the PSR indicated by LCROSS observations depends on the type of ice storing the volatiles. We consider two cases based on types of ice that would be stable in the PSR regolith: condensates and clathrates. Condensates are volatiles condensed onto regolith grains, while clathrates are volatiles trapped in water cages. If the volatiles are stored as condensates, then each species is released according to its volatility temperature, as assumed in refs. [3,8]. Volatility temperature is defined as the temperature at which pure solid evaporates from the surface to vacuum at a rate of 1 mm/billion years assuming a bulk density of 1 g/cm³ [9] as calculated by[8] using[10] (See supplementary Table S1). The long-term stability of each species depends on how the temperature varies diurnally with depth[11]. Thermal modeling shows that temperatures are stable below ~0.2 m depth[12]. The LCROSS impactor was estimated to have excavated material from 1 to 3 m deep in the PSR[13], so the volatiles observed in the plume originated below the depth of thermal stability. Additionally, Cabeus is one of the coldest PSRs, with diurnal variation in surface temperature between 38.7 and 46.7 K and subsurface temperatures estimated to be 38 K[11]. This means that most condensed volatiles in this PSR should remain stable long-term on the surface and at depth. We use regolith volatile abundances estimated by[8] based on the LCROSS plume composition, adjusting CO and $H_2$ based on our reanalysis of LAMP observations (See Supplementary Table S1). Elemental ratios for volatiles sampled by LCROSS, assuming they were condensed in the regolith, are identified in Fig. 1 as Condensates (See Supplementary Table S2).

If the volatiles are stored in clathrates, then the measured plume composition is a reasonable representation of the volatile abundance in the regolith. This is because all volatiles trapped in clathrates are released together when clathrates become destabilized. The LCROSS elemental ratios for volatiles stored as clathrates are identified in Fig. 1 as Clathrates (See Supplementary Table S2). We show in Fig. 2 that clathrates are stable at the temperatures and pressures beneath the surface in the Cabeus PSR.

### Volatile sources

The potential source or combination of sources for volatiles sampled by LCROSS will depend on the timing for volatile delivery. Cabeus crater is estimated to be 3.5 billion years old[14], providing an upper limit for the age of these volatiles. A lower limit comes from modeling the influence of impact gardening on ice deposits. Based on the abundance of ice detected by LCROSS and the depth probed by the impactor, the volatiles sampled should be from more than 1 Gya[15]. Although volcanic outgassing was most active more than 3 Gya, activity continued until at least 1 Gya[16]. In fact, several lines of evidence point to continuing release of volatiles from the Moon's interior[17] demonstrating that volatiles of a volcanic-type origin cannot be ruled out based on deposit age. Throughout its history the Moon has been subject to impacts,

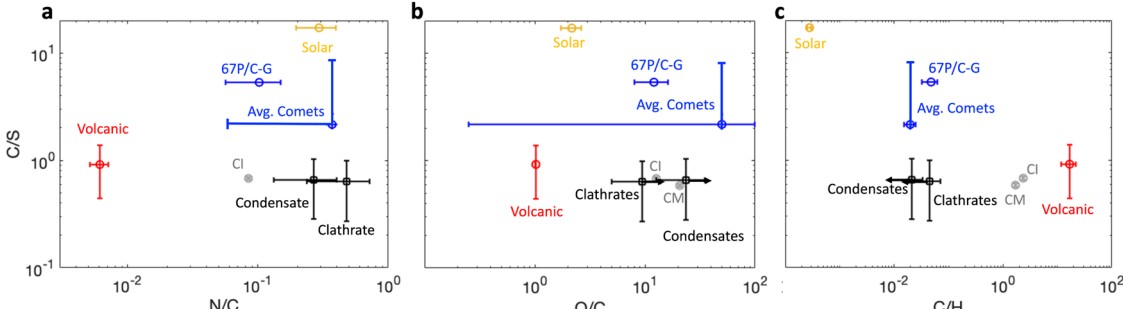

**Fig. 1 Elemental composition of the Lunar regolith in the top 1–3 m of the Cabeus Crater Permanently Shaded Region sampled by the Lunar CRater Observation and Sensing Satellite compared to the elemental composition of possible sources.** The regolith elemental composition (black squares) of (**a**) N/C, (**b**) O/C, and (**c**) C/H compared to C/S is determined based on the assumption that the volatiles are either stored as Clathrates or are condensed onto the regolith as Condensates. All sources are identified by name in the figures next to their symbol. Uncertainties are extrapolated from reported measurements according to standard methods. Note that no single source exactly matches all of the elemental ratios. Data are provided in Supplementary Table S2.

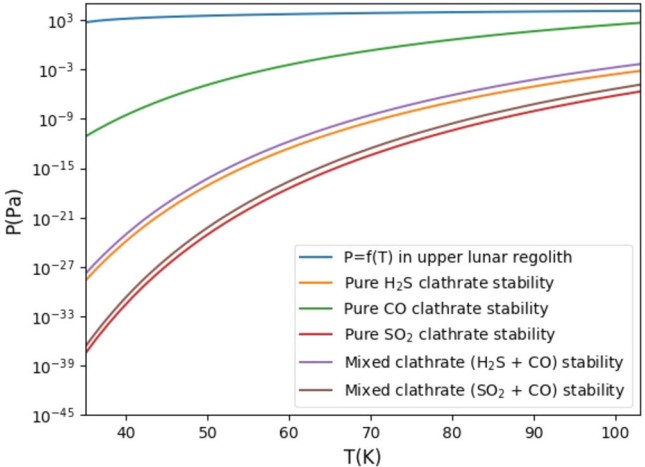

**Fig. 2 Stability curves for clathrates stored in the permanently shaded regions.** Clathrates are stable above and to the left of the curve. Comparison of (top blue line) the pressure-temperature profile, or $P = f(T)$, in the upper lunar regolith (0.2–5 m) to stability curves for clathrates with $SO_2$, $H_2S$, CO, and mixtures of these species as noted in the legend[31,34,35]. The $P = f(T)$ profile is calculated based on a temperature profile extrapolated from ref. [12] and pressure based on a 1.66 g/cm³ lunar regolith[36]. Mixed clathrate stability curves are based on clathrates formed from gas mixture of CO + $SO_2$ or $H_2S$ (with a cometary C/S from ref. [21]); such clathrate is dominated by $SO_2$ or $H_2S$. Because the $P = f(T)$ for the lunar regolith falls in the area above and to the left of all stability curves, the regolith is within the clathrate stability domain.

with the largest fluxes predating the formation of Cabeus, between 3.5 and 4.6 Gya[18]. However, impacts by comets and meteorites have continued since that time at a lower rate. Comets and chondrites in the form of asteroid impactors and micrometeoroids[19] are also a reasonable volatile source. Finally, water molecules can form through surface chemistry initiated by solar wind protons and travel to the PSRs[15].

In Fig. 1 we compare the elemental composition for the LCROSS observations with potential volatile sources (See Supplementary Table S2). Comet composition is based on coma measurements of sublimated ices, and varies significantly. However, refractory material in comet nuclei is likely chondritic in composition[20], so comet impacts would provide a combination of material with what we designate as cometary, as well as chondritic composition. We provide the composition for the coma of 67 P/Churyumov-Gerasimenko (67 P/C-G), using the best effort to date at determining elemental composition with *Rosetta* observations[21]. We also illustrate average or extreme values based on coma observations from several other comets[21]. The C/S measured in comets ranges between 2.2 and 8.0 when sulfur-bearing species have been detected. N/C in comets ranges between 0.06 and 0.37. Note that the nitrogen inventory for these comets does not include $N_2$, which is difficult to measure remotely. In 67 P/C-G, $N_2$ contributed ~17% of the total nitrogen inventory in the coma. The volcanic composition is from[16] with N/C from [22].

**Source mixtures**. As Fig. 1 shows, no source is a perfect fit for the LCROSS measurements. Volcanic sources and chondrites provide the right amount of sulfur, but do not provide sufficient hydrogen and nitrogen. Volcanic sources are also deficient in oxygen. Comets provide sufficient hydrogen, carbon, and nitrogen, but are depleted in sulfur—even when considering the most extreme value. Solar wind only contributes hydrogen and oxygen (see

Supplementary Discussion). We developed a model to determine if a mixture of sources can match the LCROSS observations, and found that no combination was able to match all four elemental ratios within the uncertainties of the LCROSS measurements – even when taking into account the uncertainties for the sources (see Supplementary Discussion). The main limitation is fitting both the C/S and the N/C ratios observed by LCROSS. The two sources with sufficient sulfur to match C/S, volcanoes and chondrites, are too depleted in nitrogen and hydrogen for any cometary contribution to provide agreement with N/C and still match C/S. This is the case even using the maximum N/C and the minimum C/S for comets. The best fit is provided by 100% comets, which agrees with all ratios except for C/S.

To improve our constraints on the source, or mixture of sources, we consider processes that could fractionate elemental ratios between delivery of the source volatiles to the lunar surface and observation in the LCROSS plume, including volcanic atmospheric processes, impact processes, clathrate formation, and cycles of sublimation and recondensation. Because these processes are complex and difficult to accurately quantify, we determine whether the LCROSS observations represent upper or lower limits for the elemental ratios and summarize the results in Table 1.

**Volcanic atmosphere fractionation**. Volcanic sulfur is thought to be released as $S_2$, which could rapidly be lost to the surface as solid elemental sulfur or aerosols before reaching a cold trap[23]. This would result in a higher C/S ratio in the PSR compared to the source, so the observed C/S is an upper limit for volcanic C/S. This creates a challenge for explaining the LCROSS C/S as volcanic in origin, because volcanic C/S would need to be much lower than C/S in the LCROSS plume to provide sufficient sulfur to explain the observations.

The relative abundances of elements in volcanic gas can also be changed by the escape of molecules from the top of the atmosphere. Unfortunately, loss rates depend on a wide range of complex parameters that are not well constrained[24], making it difficult to quantify how much elemental ratios can fractionate as a result of escape. However, we can estimate upper and lower limits for LCROSS measurements compared to the sources based on the relative masses of the dominant species for each element. Escape from a volcanic atmosphere would be dominated by H and $H_2$[23,24] that either originated in the volcanic gas as $H_2$, or was produced by dissociation of water molecules. This would increase the C/H of the volatiles in the PSR, making the observations an upper limit for the source ratio. Atomic oxygen and OH produced by water dissociation could also be lost, making O/C in the PSR a lower limit compared to the source. Any nitrogen present would be in the form of either $N_2$ or $NH_3$, which are either the same mass as or lighter than volcanic carbon-bearing molecules CO and $CO_2$. This means that the N/C in the PSR is a lower limit for N/C in a volcanic source when considering atmospheric escape. Because volcanic N/C is drastically lower than the LCROSS observations, escape does not provide a mechanism allowing for volcanic gas to be the source of nitrogen in the Cabeus PSR.

Although escape of hydrogen and oxygen leads to limits that provide worse agreement between a volcanic source and the LCROSS observations, water produced by solar wind surface chemistry would decrease C/H and increase O/C over time by adding water to the PSR[25], canceling out escape fractionation. These ratios would allow for a combination of volcanic and solar wind sources. However, the measured N/C ratio disagrees with volcanic source composition, even accounting for processes that change elemental ratios in a volcanically produced atmosphere,

**Table 1 Model constraints and results for determining the possible sources for the LCROSS plume based on understanding of fractionation processes.**

| Ratio | No fractionation | Volcanic atmosphere processes | Impact and escape | Clathrate formation | Sublimation and recondensation | Escape, sublimation and recondensation |
|---|---|---|---|---|---|---|
| C/S | Fit to observations | Upper limit | Lower limit | Lower limit | Lower limit | Lower limit |
| N/C | Fit to observations | Lower limit | Lower limit | Unconstrained | Lower limit | Lower limit |
| O/C | Fit to observations | Lower limit | Lower limit | <6.75 | Upper limit | Unconstrained |
| C/H | Fit to observations | Upper limit | Upper limit | >0.07 | Lower limit | Unconstrained |
| N/S | n/a | n/a | Lower limit | n/a | Lower limit | Lower limit |
| S/O | n/a | n/a | Upper limit | n/a | Lower limit | Unconstrained |
| S/H | n/a | n/a | Upper limit | n/a | Lower limit | Unconstrained |
| O/H | n/a | n/a | Upper limit | n/a | Lower limit | Constrained by solar wind input |
| Results | No good fit | No good fit | Comets and Solar Wind | No good fit | 30–45% Comets 55–70% Chondrites | Comets and Chondrites |

In the case of no fractionation, the model was determined to be a good fit if the modeled ratios were within errors of the LCROSS observations and the uncertainties of the sources. The N/S, S/O, S/H, and O/H constraints for impact and escape were not used in the impact and escape modeling, but were used when determining constraints for the final column that combined impact and escape with sublimation and condensation.

conclusively demonstrating that the volatiles sampled by LCROSS are not from a volcanic source.

**Fractionation of impact material.** Next, we consider fractionation of volatiles delivered by impacts of comets, asteroids, and micrometeoroids. The elemental ratios can be fractionated by impact loss and by escape during transport to cold traps. The total percentage of volatiles retained after impact depends on the impact velocity and angle[26]. Volatiles lost to space escape rapidly as part of the outward flow of the impact plume. Fractionation is similar to hydrodynamic escape, with preferential loss of lighter species. However, light species flow outward rapidly enough to drag heavier species with them (e.g., ref. [27]). Additional loss to space could occur by escape during subsequent transport to cold traps over several Earth days[28]. Fractionation can be estimated in the same way as with the volcanic atmosphere, assuming that lighter species are removed at a faster rate than heavier species. Hydrogen would primarily be in light molecules like H, $H_2$, and water making the C/H in the PSR an upper limit compared to C/H of the source. Loss of oxygen and OH would make O/C in the PSR a lower limit compared to the source. According to simulations of impact chemistry of comets[29] and chondrites[30], nitrogen in an impact plume would primarily be in the form of $N_2$ with some $NH_3$ present, while carbon and sulfur are found in heavier molecules like CO, $CO_2$, $H_2S$, $SO_2$, and OCS. As with the volcanically produced atmosphere, N/C in the LCROSS observations is a lower limit compared to the source. We also note that LCROSS and LAMP did not have the ability to detect $N_2$, which is expected to be produced in impact plumes. The N/C in the LCROSS plume may have been higher than observed, arguing further that the observation is a lower limit compared to the source. Finally, although the loss of hydrogen would be greater than the loss of oxygen, making O/H an upper limit. The masses for carbon-bearing species are generally lighter than sulfur-bearing species, suggesting that C/S in the LCROSS observations is a lower limit compared to the source. We applied our model again using these constraints (see Table 1) and found that only cometary ices, with some contribution from solar wind-produced water, can explain all four elemental ratios.

**Clathrate formation.** During the cooling of an impact plume, clathrates can form with entrapped mixtures different from the coexisting gases. In this case, the entrapped mixture will be enriched in $H_2S$ and $SO_2$, and depleted in CO compared to the initial mixture because $H_2S$ and $SO_2$ have a higher propensity for trapping compared to CO at low pressure conditions[31]. If insufficient water is available to trap all of the CO, $H_2S$ and $SO_2$ present in the gas, C/S in the clathrates is lower than in the source. Ammonia is not trapped in clathrates, but would form ammonia hydrates at temperatures between 80 and 100 K, or condense as pure ammonia frost at temperatures below 80 K. If not all of the CO is trapped, but all of the $NH_3$ ends up in the PSR, the N/C observed by LCROSS is an upper limit compared to the source. In this case, either comets or chondrites could agree with the C/S and N/C. However, based on the water to CO ratio in clathrates the C/H ratio for volatiles trapped in clathrates must be higher than 0.09 and the O/C ratio must be lower than 6.75 if not enough water was available for all of the CO to be trapped[31]. Although the LCROSS O/C is greater than this limit, this could be explained by additional water supplied by the solar wind.

Because clathrate formation would not occur in isolation, we considered a combination of clathrates and escape. In this scenario, we modeled a combination of sources assuming that the LCROSS C/S and O/C are lower limits based on clathrate formation processes and escape, that C/H is an upper limit based on escape, and ignoring N/C because of the competing influences of clathrate formation and escape. We found that a combination of cometary and solar wind sources fits these constraints, but that the modeled O/C is too high to support clathrate formation even accounting for a solar wind water source. Therefore, it is unlikely that ices that formed as clathrates can explain the LCROSS observations.

**Sublimation and recondensation.** Finally, we consider how a cycle of sublimation and recondensation of volatiles could fractionate the elemental ratios. As volatiles are transported to the PSR, they could condense to the surface at night and sublimate during the day. A similar cycle could also take place within a PSR if diurnal temperatures vary enough to cause sublimation of some species depending on their volatility. The temperatures in the Cabeus PSR are very low and not likely to cause diurnal variations, but volatiles in this PSR could have been influenced by these processes before being trapped. Additionally, recondensation could occur within the Cabeus PSR when volatiles are released through impact gardening. This cycle would increase

the abundance of water relative to other species observed in the LCROSS plume that have lower volatility temperatures (See Supplementary Table S1). It would also increase the abundance of $NH_3$, $H_2S$, and $SO_2$ relative to CO and $N_2$. This means that C/S, and C/H in the PSR are lower limits compared to the source, while O/C is an upper limit. Although $NH_3$ increases relative to CO, impacts are more likely to produce $N_2$ than $NH_3$. Additionally, the $N_2$ and CO volatilities are similar so this process removes twice as many nitrogen atoms than carbon atoms for each molecule lost. Therefore, N/C is a lower limit. We modeled the source contributions with these constraints and found that a combination of comets, chondrites, and solar wind was possible. To narrow the possibilities further we add four more constraints shown in Table 1. Because sulfur-bearing species are lost more easily than water, S/O and S/H are lower limits. Additionally, several oxygen-bearing species are more likely to be lost than the main hydrogen-bearing species, indicating that O/H is also a lower limit. Including these constraints limits the possible combination of source volatiles to 30–45% cometary and 55–70% chondrites with no solar wind contributions. Finally, we consider the combination of loss to space and a cycle of sublimation and recondensation. By comparing the columns for these two cases in Table 1, we can see that the constraints for several ratios offset each other. Because of this, the only reliable constraints are C/S, N/C, N/S, and O/H. These constraints allow for any combination of comets and chondrites with no water provided by solar wind.

## Discussion

Because no combination of known sources is able to match the large abundances of both sulfur and nitrogen compared to carbon measured by LCROSS we had to consider fractionation of the elements between delivery of volatiles to the surface of the Moon and trapping in the PSRs. The large nitrogen abundance allows us to rule out a volcanic atmosphere as a source for any of the volatiles even accounting for the fractionating process. The fractionation of the elemental ratios by loss of volatiles to space and a cycle of sublimation and recondensation allows for a combination of cometary and chondritic material for the volatiles observed by LCROSS. Recognizing that the refractory material in comets is likely chondritic in composition, comets alone are a reasonable source and are likely the primary source of these volatiles.

Measuring the elemental composition and the isotope ratios of the five elements evaluated in this study as a function of depth within the Cabeus PSR would provide constraints on the relative contribution of the solar wind to the water sampled, as well as details about the impactors. Because the isotope ratios of each source differ enough to serve as a tracer of the source, mapping them with depth would allow us to map out the composition of impactors as a function of time. Additionally, noble gas abundances and their isotope ratios are extremely valuable for tracing the sources of volatiles delivered to the Moon (e.g., ref. [32]). As humans prepare to return to the Moon[33], we have an unprecedented opportunity to make such measurements in Cabeus and other PSRs. It is essential that future lunar missions have a plan to characterize the elemental and isotopic composition of lunar volatiles as a function of depth as they are accessed prior to converting volatiles to resources needed for human exploration.

## Data availability

The authors declare that all data supporting the findings of this study are available within the paper and its supplementary information.

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

## Acknowledgements

K.E.M. and D.M.H. acknowledge support by the NASA LRO project through LAMP Subcontract A99129JD. K.D.R. and L.O.M. acknowledge support by the NASA LRO LAMP project under contract NNG05EC87C. K.E.M. and A.L.K. acknowledge support by NASA RDAP grant 80NSSC19K1306. O.M. and A.B. acknowledge support from CNES. The project leading to this publication has received funding from the Excellence Initiative of Aix-Marseille Université - A*Midex, a French "Investissements d'Avenir programme" AMX-21-IET-018.

## Author contributions

K.M. and O.M. conceptualized the study and developed the methodology. K.M. conducted the data analysis, developed the model, and led the writing and visualization of the results. D.H. led the data analysis for the LAMP observations with assistance from K.R. and L.M. A.B. and A.L.-K. assisted with the data analysis and modeling.

## Competing interests

The authors declare no competing interests.
