## [Peer Review File · Nature Communications]

REVIEWER COMMENTS

Reviewer #1 (Remarks to the Author):

This work by Mandt et al. on the lunar volatiles of the PSRs as measured by LCROSS is significant and has strong methodology and could merit publication in this journal. However, the authors must consider the contribution of lunar outgassings, not necessarily related to traditional volcanism, in the list of possible gas origins, and the impact of these materials on their results. They should see the following papers by Crotts:
<https://iopscience.iop.org/article/10.1086/591634/meta>
<https://iopscience.iop.org/article/10.1088/0004-637X/707/2/1506/meta>
<https://www.tandfonline.com/doi/pdf/10.1080/21672857.2012.11519702>

The compositions of the gases released have not been directly measured, and may generally be expected to be similar to other volcanic gases. However, ALSEP measurements indicated the possibility that less traditionally "volcanic" gases are released in outgassings, such as CH₄, Ar, NH₃, in addition to H₂O, CO, CO₂, N₂ and O₂ (Paper I above). These considerations may have a significant impact on the paper's results.

Fig. 2 – Can you please show the stability region for the Moon on this plot to help place the calculations in context?

Line 104 – check "sh"

Line 223 – water could also be supplied from the interior by late-stage lunar outgassing, or Transient Lunar Phenomena, see Crotts et al. paper 1 above. For that matter, there may be other gases from these events that could affect the gas proportions, and they have not been addressed here.

Line 278 – I wholeheartedly agree that these volatiles must be carefully studied before they are modified by exploration.

No other needed corrections to the text or organization were noted.

Reviewer #2 (Remarks to the Author):

This paper analyzes the nature of the volatiles released during the LCROSS impact. In particular, the authors want to know if the volatiles are of exogenic or endogenic origin. By thoroughly comparing the sampled volatiles' chemical composition with the chemical composition of various possible sources (including modifications thereof over time), the authors rule out an endogenic origin and show that the most likely source candidates for the measured impact volatiles are comets.

The work is both highly original and highly timely. With several space agencies planning to return to the Moon within the next decade, the potential of such analyses (determining the origin of lunar volatiles) should be determined well in advance, with volatiles being of high interest not only due to their usability but also due to the scientific questions they can help answer (e.g., origin of water on the Moon). This observation presents the opportunity to test such analyses now and demonstrate that from measurements of today's lunar volatiles, some strong points for/against certain sources thereof can be made.

Whereas a definite answer to the question of the lunar volatile source could not be made, the analysis makes full use of the available possibilities, thus presenting the best effort. The conclusion is sound, credible, and well comprehensible. The methodology is sound and of high quality.

I could not find any flaws. The steps to reach certain conclusions, however, seem to me to be not sufficiently presented. The conclusions I had difficulties reproducing are mentioned in the points given below.

There are a few text-related points that I would like to give below:

I31: Not only between 'the source and storage' but 'the source and release', no? The former sounds as if once the material is at the PSR it is not modified further (or make clear that storage means not the beginning of storage but the whole duration of storage)

I35: instead of what?

I53: Which condensed volatiles? The ones presented in table S1? Or only CO and H₂? Also, even though < 40K is very low, do H₂, CO, and N₂ really condense at the PSR temperatures? Is this something Berzhony shows? Because I wouldn't have expected these volatiles to condense even at temperatures as low as ~40K...

I67: Is this directly what was measured? 'Un-processed'?

I71: Maybe you can mention in the caption where wrt the curves volatiles are stable (above and to the left).

I87: sh -> so

I97: Can you please provide a reference for this?

I101: from several *other* comets?

I109: But you wouldn't *expect* it to be a perfect fit, right? Because, as you discuss in length, even if the original material was a perfect match, since its delivery to the surface it will have been chemically altered to some extent.

I158: I am not entirely convinced of this statement when it comes to C/S. It seems that C/S could be decreased and still not disagree with the volcanic C/S error bars (there is still room for overlap), no?

I189: In all of this I find the O/C considerations hard to follow.

I203: In this scenario?

I204: Is O/C being a lower limit also based on the clathrate formation process? I am not sure I understand how O/C is a lower limit.

I205: This is counteracted by clathrate formation enriching the entrapped mixture in H₂S and depleting it in CO though, no?

I222: How do you determine if N/C is a lower limit if NH₃ increases relative to CO and N₂?

I228: This is the last column in table 1, i.e., with added escape, right? And this escape would have occurred during the initial deposition? Because even at 400K sublimation at the Moon does not result in escape due to the Moon's high gravity in comparison (with the exception of H₂). (Or are you considering sputtering?)

I231: This seems a little short and I have difficulties reaching the conclusions you reach on your own. Can you please elaborate on this?

I234: You didn't use N/S...O/H given for impact and escape, right? I suggest mentioning that these values are presented here but were not used in the analysis.

I234: There are some N/S...O/H limits I cannot retrace:

O/H being an upper limit in 'Impact and Escape'

S/O, S/H, and O/H being a lower limit in 'Sublimation and Recondensation'

(btw. there is inconsistency in capitalization in the table header)

I234: Where does the >0.07 for C/H in 'Clathrate Formation' come from?

I259: The whole Summary and Conclusion section is very well written! It presents a clear and concise summary of the paper and a well-thought-through outlook!

I274: Is 50km at 60s? Because I24 mentions >100 km

I282: 200K appears very low. I was expecting something on the order of several thousand K. Or is this for some time after the impact when the material has cooled down? This wouldn't agree with the source rate being constant throughout the first 150s, though...

I284: CO source rate? Total source rate?

I287: Is 90s the time of the second peak?

I290: Where does the 5500kg come from? Is that also calculated for 150s?

I291: The water mass is missing, no? How do you get 5%?

I300: they -> then

I302: I am not sure I get the concept of how the material is released. Is this assuming that the impact first excavates some material and then the remainder is released by remnant heat? Or is the initial impact plume existing over a long period of time?

I312: I'm not sure I understand the first two columns. Is column 1 what you measured (Figure 1)? You say in the clathrate scenario the plume composition represents the regolith composition. So I am confused that 'regolith' is mentioned only in the second column.

And is column 2 column 1 (measurement) but corrected for volatility? Or is that from literature?

I312: Why don't the two first columns add up to 100%?

Overall I thoroughly enjoyed reading this paper. It is of high quality, presenting exciting research, and throughout well written.

best,
Audrey

Reviewer #3 (Remarks to the Author):

The manuscript "An Exogenic Origin for the Volatiles Sampled by the LCROSS Impact" gives an overview of the various volatiles and possible origins (and evolution) of those volatiles in relation to the LCROSS impact experiment. This is an interesting perspective on such volatiles and how the LCROSS experiment data should be reanalyzed. I do advise very minor revision of the manuscript, though mostly through the addition of information or clarification on some statements made.

Abstract: abstract should not include references. Please double check the author guidelines.

Section 3: This section may benefit with a table or reference to soil abundances of elements/volatiles (e.g., H, S, CO, etc.).

Line 103 - 105: Sentence unclear, what is sh?

Line 216: Please add reference to the ammonia/ammonia temperatures provided.

Maybe an added section on the LCROSS impact that penetrated the lunar surface at a certain depth and velocity, which would have heated the area immediately (I believe up to 1000 K from Gladstone et al. 2010) which in turn could liberate large amounts of CO during this impact-pyrolysis...

Reference #37: Please include full reference for "Fray and Schmitt"

Dear Reviewers,

We would like to thank the Reviewers for the time and care you have put into this manuscript. Their feedback made several valuable points that we have addressed in our revision.

Below we address each of the reviewer comments in indented, blue text with relevant notable text added to the manuscript **bolded**. Revisions to the manuscript are also indicated in **blue bolded text** in the annotated version.

REVIEWER COMMENTS

Reviewer #1 (Remarks to the Author):

This work by Mandt et al. on the lunar volatiles of the PSRs as measured by LCROSS is significant and has strong methodology and could merit publication in this journal. However, the authors must consider the contribution of lunar outgasings, not necessarily related to traditional volcanism, in the list of possible gas origins, and the impact of these materials on their results. They should see the following papers by Crofts:

<https://iopscience.iop.org/article/10.1086/591634/meta>

<https://iopscience.iop.org/article/10.1088/0004-637X/707/2/1506/meta>

<https://www.tandfonline.com/doi/pdf/10.1080/21672857.2012.11519702>

The compositions of the gases released have not been directly measured, and may generally be expected to be similar to other volcanic gases. However, ALSEP measurements indicated the possibility that less traditionally "volcanic" gases are released in outgasings, such as CH₄, Ar, NH₃, in addition to H₂O, CO, CO₂, N₂ and O₂ (Paper I above). These considerations may have a significant impact on the paper's results.

We thank you for bringing these important papers to our attention. They are definitely directly relevant to this study. The main concern in excluding this source would be if the N/C of recently outgassed material is significantly larger than our N/C for volcanic gas. After reviewing the papers above and the publications outlining Apollo and LADEE measurements of the exosphere, we conclude that the most reasonable prediction for lunar outgassing composition is traditionally volcanic. We agree that this needs to be outlined in the paper and have added the following to the discussion of volcanic gas in section 3: **“Although volcanic outgassing was most active more than 3 Gya, activity continued until at least 1 Gya (21). In fact, several lines of evidence point to continuing release of volatiles from the Moon’s interior (22) demonstrating that volatiles of a volcanic-type origin cannot be ruled out based on deposit age.”** We have also realized based on this feedback that more details are needed on our source composition determinations. We have, therefore, added a section to the Supplemental Materials that explains the source compositions in more detail. The portion relevant to outgassing states: **“The relative abundances of CO, H₂O, H₂, OH, and S in volcanic gas was estimated by (10) based on a review of Apollo sample composition measurements. Nitrogen abundance in lunar volcanic gas was determined by (11) based on lunar sample composition, which suggests that N/C in**

lunar volcanic gas is an order of magnitude larger than terrestrial volcanic gas. We also reviewed lunar exosphere measurements to ensure that our source composition included potential volatiles provided by ongoing interior outgassing on the Moon through events described as Transient Lunar Phenomena (12). The lunar exosphere is primarily made up of helium, neon, and argon (13). Trace amounts of CH₄ and NH₃ have been detected, and are interpreted to be derived from solar wind carbon and nitrogen (14,15). Although the argon isotope, ⁴⁰Ar, comes from the interior of the Moon through radiogenic decay of heavier elements, no carbon-, nitrogen-, oxygen-, or sulfur-bearing species have been clearly connected to related internal diffusion and outgassing processes (16). Therefore, we assume that TLP composition is similar enough to volcanic gas composition determined by the studies outlined in (10,11). However, as noted by (12), studying the composition of TLPs is essential now in their pristine state, before extensive robotic and human exploration of the Moon introduces large amounts of anthropogenic gas to the exosphere and TLPs.”

Fig. 2 – Can you please show the stability region for the Moon on this plot to help place the calculations in context?

The stability region for the Moon is shown as the blue line, $P=f(T)$ in upper lunar regolith.

Line 104 – check “sh”

This has been corrected.

Line 223 – water could also be supplied from the interior by late-stage lunar outgassing, or Transient Lunar Phenomena, see Crofts et al. paper 1 above. For that matter, there may be other gases from these events that could affect the gas proportions, and they have not been addressed here.

We agree that this potential source needs to be included and have made the revisions outlined above.

Line 278 – I wholeheartedly agree that these volatiles must be carefully studied before they are modified by exploration.

We are hopeful that our study can amplify this issue as exploration efforts grow.

No other needed corrections to the text or organization were noted.

Reviewer #2 (Remarks to the Author):

This paper analyzes the nature of the volatiles released during the LCROSS impact. In particular, the authors want to know if the volatiles are of exogenic or endogenic origin. By thoroughly comparing the sampled volatiles' chemical composition with the chemical composition of various

possible sources (including modifications thereof over time), the authors rule out an endogenic origin and show that the most likely source candidates for the measured impact volatiles are comets.

The work is both highly original and highly timely. With several space agencies planning to return to the Moon within the next decade, the potential of such analyses (determining the origin of lunar volatiles) should be determined well in advance, with volatiles being of high interest not only due to their usability but also due to the scientific questions they can help answer (e.g., origin of water on the Moon). This observation presents the opportunity to test such analyses now and demonstrate that from measurements of today's lunar volatiles, some strong points for/against certain sources thereof can be made.

Whereas a definite answer to the question of the lunar volatile source could not be made, the analysis makes full use of the available possibilities, thus presenting the best effort. The conclusion is sound, credible, and well comprehensible. The methodology is sound and of high quality.

We thank you for these comments.

I could not find any flaws. The steps to reach certain conclusions, however, seem to me to be not sufficiently presented. The conclusions I had difficulties reproducing are mentioned in the points given below.

There are a few text-related points that I would like to give below:

I31: Not only between 'the source and storage' but 'the source and release', no? The former sounds as if once the material is at the PSR it is not modified further (or make clear that storage means not the beginning of storage but the whole duration of storage)

We have changed this to say "...between the source, storage in the PSR, and release into the plume."

I35: instead of what?

We have revised this to say "instead of using molecular composition..."

I53: Which condensed volatiles? The ones presented in table S1? Or only CO and H₂? Also, even though < 40K is very low, do H₂, CO, and N₂ really condense at the PSR temperatures? Is this something Berzhony shows? Because I wouldn't have expected these volatiles to condense even at temperatures as low as ~40K...

Yes, these three volatiles would not condense at ~40 K. We have added "most" to this sentence to recognize that.

l67: Is this directly what was measured? 'Un-processed'?

We have added measured here.

l71: Maybe you can mention in the caption where wrt the curves volatiles are stable (above and to the left).

This has been added.

l87: sh -> so

This has been corrected.

l97: Can you please provide a reference for this?

We have added Joswiak et al. (2017) for this.

l101: from several *other* comets?

This change has been made.

l109: But you wouldn't *expect* it to be a perfect fit, right? Because, as you discuss in length, even if the original material was a perfect match, since its delivery to the surface it will have been chemically altered to some extent.

Yes. Our initial hypothesis was that using elemental ratios instead of molecular composition would provide a source or combination of sources that was a perfect fit. This first step outlined here was intended to demonstrate that this hypothesis is false and that fractionation must be considered.

l158: I am not entirely convinced of this statement when it comes to C/S. It seems that C/S could be decreased and still not disagree with the volcanic C/S error bars (there is still room for overlap), no?

We agree that N/C is really the decisive ratio for a volcanic source and have removed C/S from this statement.

l189: In all of this I find the O/C considerations hard to follow.

l203: In this scenario?

l204: Is O/C being a lower limit also based on the clathrate formation process? I am not sure I understand how O/C is a lower limit.

l205: This is counteracted by clathrate formation enriching the entrapped mixture in H₂S and

depleting it in CO though, no?

What makes it most confusing is probably that we did not effectively separate the two scenarios we considered in this section: clathrates only, and clathrates combined with escape. We have separated the clathrates combined with escape scenario into its own paragraph with an introductory sentence to make it stand out better, which hopefully helps.

I222: How do you determine if N/C is a lower limit if NH₃ increases relative to CO and N₂?

This is a good point and identifies where we need to provide a better explanation. We have added the following: “Although NH₃ increases relative to CO, impacts are more likely to produce N₂ than NH₃. Additionally, the N₂ and CO volatilities are similar so this process removes twice as many nitrogen atoms than carbon atoms for each molecule lost. Therefore, N/C is a lower limit.”

I228: This is the last column in table 1, i.e., with added escape, right? And this escape would have occurred during the initial deposition? Because even at 400K sublimation at the Moon does not result in escape due to the Moon's high gravity in comparison (with the exception of H₂). (Or are you considering sputtering?)

Yes, in this case of combining processes, we assume that the escape being evaluated takes place during initial deposition.

I231: This seems a little short and I have difficulties reaching the conclusions you reach on your own. Can you please elaborate on this?

We have revised the explanation to say the following, which may be clearer than the previous description: “By comparing the columns for these two cases in Table 1, we can see that the constraints for several ratios offset each other. Because of this, the only reliable constraints are C/S, N/C, N/S and O/H.”

I234: You didn't use N/S...O/H given for impact and escape, right? I suggest mentioning that these values are presented here but were not used in the analysis.

Yes, this is correct. We have added this explanation to the Table 1 header.

I234: There are some N/S...O/H limits I cannot retrace:
O/H being an upper limit in 'Impact and Escape'
S/O, S/H, and O/H being a lower limit in 'Sublimation and Recondensation'
(btw. there is inconsistency in capitalization in the table header)

We have added explanations to the “Impact and Escape” and “Sublimation and Recondensation” sections for these ratios and have corrected the table header.

l234: Where does the >0.07 for C/H in 'Clathrate Formation' come from?

The reason is the same as for O/C, based on the water molecules in the clathrate cage compared to the molecules trapped. We explained the O/C limit in section 4.3, but did not explain C/H. We have added this to section 4.3.

l259: The whole Summary and Conclusion section is very well written! It presents a clear and concise summary of the paper and a well-thought-through outlook!

Thank you!

l274: Is 50km at 60s? Because l24 mentions >100 km

These two references are to different parameters of the observation. Distance from impact to shell @30s is >100 km. In the second reference, the distance from the LAMP field of view to the shell was > 50 km, as described in the text. We have updated the first reference to clarify the difference in parameters being described.

l282: 200K appears very low. I was expecting something on the order of several thousand K. Or is this for some time after the impact when the material has cooled down? This wouldn't agree with the source rate being constant throughout the first 150s, though...

The model assumes that the CO is subliming off of the material lofted into sunlight. In this case 200K is a reasonable assumption. Although there is probably material subliming off of the hot crater, which was closer to 1000 K, we assume that this material left very rapidly and is not included in the portion of the plume LAMP is observing. We have revised this section to clarify this.

l284: CO source rate? Total source rate?

Because the model treats each species separately, we have updated this to say "CO source rate."

l287: Is 90s the time of the second peak?

Yes, 90s is the time of the second peak. We have added this to the text.

l290: Where does the 5500kg come from? Is that also calculated for 150s?

The ~ 5500 kg is based on different estimates from Colaprete et al. (2010) and Schultz et al. (2010). We have added these references.

I291: The water mass is missing, no? How do you get 5%?

We have added the water mass to this.

I300: they -> then

This has been corrected.

I302: I am not sure I get the concept of how the material is released. Is this assuming that the impact first excavates some material and then the remainder is released by remnant heat? Or is the initial impact plume existing over a long period of time?

This is a good question. We have added the following to this section to provide more details on this concept: "In this case we must estimate the abundance of the volatile in the regolith based on the total volume of regolith from which the volatile was released. The heating of the surface by the impact will result in temperatures that are highest at the impact point and decrease with distance from the impact. Volatiles with a lower volatility temperature will be released from a larger volume of regolith than volatiles that have a higher volatility temperature. We determined the abundance of each volatile in the regolith using the study conducted by (10) but correcting for the abundances of CO and H₂ that were calculated as part of this work."

I312: I'm not sure I understand the first two columns. Is column 1 what you measured (Figure 1)? You say in the clathrate scenario the plume composition represents the regolith composition. So I am confused that 'regolith' is mentioned only in the second column. And is column 2 column 1 (measurement) but corrected for volatility? Or is that from literature?

We have updated the column title and table introduction to clarify this by adding: "The volatiles measured in the plume are assumed to be the same composition as in the regolith if the volatiles are stored as clathrates. This is because all volatiles are released together when clathrates destabilize. The estimates for volatiles condensed on regolith grains are based on the analysis of each species volatility temperature conducted by (8) correcting CO and H₂ to agree with LAMP observations."

I312: Why don't the two first columns add up to 100%?

The reviewer brings up a good point. These are how the values were reported in the literature, and are clearly not percent as we had assumed, but instead abundance relative to water. We have removed the percent symbol from the top of the column and revised the table description to clarify this.

Overall I thoroughly enjoyed reading this paper. It is of high quality, presenting exciting research, and throughout well written.

best,
Audrey

We thank you so much for your thorough and fair review of this manuscript and are very happy to hear that you enjoyed reading the paper.

Reviewer #3 (Remarks to the Author):

The manuscript “An Exogenic Origin for the Volatiles Sampled by the LCROSS Impact” gives an overview of the various volatiles and possible origins (and evolution) of those volatiles in relation to the LCROSS impact experiment. This is an interesting perspective on such volatiles and how the LCROSS experiment data should be reanalyzed. I do advise very minor revision of the manuscript, though mostly through the addition of information or clarification on some statements made.

Abstract: abstract should not include references. Please double check the author guidelines.

We have removed the references from the abstract.

Section 3: This section may benefit with a table or reference to soil abundances of elements/volatiles (e.g., H, S, CO, etc.).

We provide tables in the supplemental materials that lists what is known for the molecules (Table S1) and elements (Table S2) in the soil based on the LCROSS observations and for the sources.

Line 103 - 105: Sentence unclear, what is sh?

This has been corrected to “so”.

Line 216: Please add reference to the ammonia/ammonia temperatures provided.

The volatility temperatures and their references are provided in Table S1. In the case of ammonia, the reference is the same as most of the molecules and is provided in the column header as reference (17). We chose this approach to save space within the table, but are happy to add (17) after each of the temperatures in this column if needed.

Maybe an added section on the LCROSS impact that penetrated the lunar surface at a certain depth and velocity, which would have heated the area immediately (I believe up to 1000 K from Gladstone et al. 2010) which in turn could liberate large amounts of CO during this impact-pyrolysis...

In section 2 we note the depth of material that was excavated. We have added reference to this as well in section 6.1 along with reference to the temperature.

Reference #37: Please include full reference for "Fray and Schmitt"

The full reference has been added.

REVIEWERS' COMMENTS

Reviewer #1 (Remarks to the Author):

Dear Editor,

The revised manuscript, An Exogenic Origin for the Volatiles... by Mandt et al. has addressed the concerns of this reviewer adequately and should be published. The paper is well organized and has compelling science that will lead to new advances for our understanding of lunar volatiles.

There is just one suggestion - Check the new sentence structure on Line 223.

Thank you - Reviewer 1

Dear Editor and Reviewers,

We would like to thank the Editor and Reviewers for the time and care you have put into this manuscript. The feedback from the Reviewers made several valuable points that we have addressed in our revision.

Below we address the reviewer's comment in indented, blue text with relevant notable text added to the manuscript **bolded**. Revisions to the manuscript are indicated using track changes.

REVIEWER COMMENTS

Reviewer #1 (Remarks to the Author):

The revised manuscript, An Exogenic Origin for the Volatiles... by Mandt et al. has addressed the concerns of this reviewer adequately and should be published. The paper is well organized and has compelling science that will lead to new advances for our understanding of lunar volatiles.

There is just one suggestion - Check the new sentence structure on Line 223.

We have revised this sentence to say **"Although the LCROSS O/C is greater than this limit, this could be explained by additional water supplied by the solar wind."**

Best regards,
Kathy Mandt and co-authors